# Current Development and Application of Anaerobic Glycolytic Enzymes in Urothelial Cancer

**DOI:** 10.3390/ijms221910612

**Published:** 2021-09-30

**Authors:** Yi-Fang Yang, Hao-Wen Chuang, Wei-Ting Kuo, Bo-Syuan Lin, Yu-Chan Chang

**Affiliations:** 1Department of Medical Education and Research, Kaohsiung Veterans General Hospital, Kaohsiung 81362, Taiwan; yvonne845040@gmail.com; 2Department of Pathology and Laboratory Medicine, Kaohsiung Veterans General Hospital, Kaohsiung 81362, Taiwan; hwchuang1980@gmail.com; 3Institute of Oral Biology, School of Dentistry, National Yang Ming Chiao Tung University, Taipei 11221, Taiwan; 4Division of Urology, Department of Surgery, Kaohsiung Veterans General Hospital, Kaohsiung 81362, Taiwan; kiwima0817@hotmail.com; 5Institute of Clinical Medicine, National Yang Ming Chiao Tung University, Taipei 11221, Taiwan; 6Department of Biomedical Imaging and Radiological Sciences, National Yang Ming Chiao Tung University, Taipei 11221, Taiwan; a5375128@gmail.com

**Keywords:** urothelial cancer, metabolic reprogramming, glycolytic enzymes

## Abstract

Urothelial cancer is a malignant tumor with metastatic ability and high mortality. Malignant tumors of the urinary system include upper tract urothelial cancer and bladder cancer. In addition to typical genetic alterations and epigenetic modifications, metabolism-related events also occur in urothelial cancer. This metabolic reprogramming includes aberrant expression levels of genes, metabolites, and associated networks and pathways. In this review, we summarize the dysfunctions of glycolytic enzymes in urothelial cancer and discuss the relevant phenotype and signal transduction. Moreover, we describe potential prognostic factors and risks to the survival of clinical cancer patients. More importantly, based on several available databases, we explore relationships between glycolytic enzymes and genetic changes or drug responses in urothelial cancer cells. Current advances in glycolysis-based inhibitors and their combinations are also discussed. Combining all of the evidence, we indicate their potential value for further research in basic science and clinical applications.

## 1. Introduction to Urothelial Cancer

Urothelial cancer occurs in the lower urinary tract, including the bladder and urethra, as well as the upper urinary tract, including the renal pelvis and ureter [1]. Various molecular characteristics and patterns of upper tract urothelial cancer (UTUC) and bladder cancer (BC) have been identified for clinical prediction and diagnosis, respectively. Interestingly, UTUC and BC have different features, leading to different variations. Previous studies have revealed that BC has a higher mutational burden compared with other common cancer types [2]. This may be partially explained by apolipoprotein B mRNA-editing enzyme, catalytic polypeptide (APOBEC)-mediated mutagenesis. Moreover, several mutated genes and chromatin modifiers (e.g., *CREBBP*, *KMT2C*, *KMT2D*, and *KDM6A*) have been reported in BC. The main targets include the RTK/Ras/PI3K pathway (*FGFR3*, *ERBB2*, *HRAS*, and *PIK3CA*) and p53–Rb axis (e.g., *TP53*, *RB1*, *CCND1*, and *CDKN2A*) (Figure 1).

On the other hand, BC-like incidents do not appear to occur in UTUC. Although the evidence related to UTUC is limited, some events have been described. Compared with BC, *CUL3*, *BCL2L1*, and *FGFR1* have been recognized as mutation events in UTUC that can potentially be used for its classification (Figure 1). These properties may be due to UTUC-related symptoms, such as Lynch syndrome (which develops in the mesoderm-derived epithelium) and aberrant aristolochic acid (AA) production [3,4]. *HRAS*, *ERBB2*, *TP53*, and *RB1* can be used as molecular markers in both types of urothelial cancer because their actions differ between normal tissues and primary tumors [5]. In addition, to comprehensively characterize genetic alterations and various expression panels of UTUC for further experiments, whole-exome sequencing, and RNA sequencing profiles need to be established and integrated through large-scale screening [6,7]. At present, the research is limited, and the number of samples is insufficient, making it difficult to achieve breakthrough progress. In addition to the above-mentioned clinical background, urothelial cancer involves many important factors, such as metabolism.

## 2. Glycolysis Pathway in UC

Glycolysis is a typical and sensitive metabolic process that plays a primary role in carbohydrate uptake/utilization, homeostasis, and metabolic reprogramming. According to the physiological function and homeostasis of the internal environment, the direction (from top to bottom or vice versa), efficiency, products, and branches of this pathway can be adjusted at any time. Glycolysis is divided into two main types, namely, aerobic and anaerobic glycolysis, and the reaction depends on the surrounding environment. The glycolytic pathway is responsible for the conversion of glucose to pyruvate and ATP production. Glycolysis first requires glucose transporters (GLUTs) to carry glucose across the cell membrane. This reaction is catalyzed by ten enzymes in the glycolytic pathway: (1) hexokinase (HK); (2) glucose-6-phosphate isomerase (GPI); (3) phosphofructokinase 1 (PFKM); (4) aldolase, aldolase, Fructose-bisphosphate (ALDO); (5) triosephosphate isomerase (TPI1); (6) glyceraldehyde-3-phosphate dehydrogenase (GAPDH); (7) phosphoglycerate kinase (PGK); (8) phosphoglycerate mutase (PGM); (9) enolase, phosphopyruvate hydratase (ENO); and (10) pyruvate kinase (PKM) (Table 1). Only 2 ATP per glucose molecule are synthesized in anaerobic glycolysis, whereas when pyruvate enters mitochondrial respiration, 36 ATP molecules per glucose molecule are produced (aerobic glycolysis). Under aerobic conditions, pyruvate enters the mitochondria, where it is completely oxidized by O_2_ to CO_2_ and H_2_O, and its chemical energy is mainly stored in the form of ATP. Pyruvate generated by aerobic glycolysis enters the tricarboxylic acid cycle (TCA) or Krebs cycle. In the absence of sufficient oxygen, pyruvate is reduced by NADH through anaerobic glycolysis or fermentation to form various products, including lactate in animals and ethanol in yeasts. On the other hand, in the glycolysis pathway, also called the Embden–Meyerhof–Parnas (EMP) pathway, the starting molecule is glucose, which is a simple and abundant sugar found in carbohydrates that provides energy to most cells. Carbohydrates synthesized in the process of photosynthesis are the main storage molecules of solar energy. After ingestion, complex carbohydrates are enzymatically hydrolyzed into monosaccharides; for example, starch is broken down into d(+)-glucose. The catabolism of glucose is the primary mechanism to meet short-term energy requirements.

In tumorigenesis, cancer cells are activated by aberrant metabolic reprogramming events. These events include changes in oxygen concentration, acidity, and various metabolites. Notably, anaerobic glycolysis is overexpressed in tumor cells and malignant sites. Otto Warburg claimed that anaerobic glycolysis and lactic acid fermentation lead to the proliferation and growth of cancer cells, which is now known as the “Warburg effect”. Scientists have successively elucidated the metabolic reprogramming of these processes in various cancer subtypes. For example, urothelial cancer cells are considered “Warburg-like”, which is characterized by a dependence on aerobic glycolysis and the expression of most glycolytic enzymes [8]. Aerobic glycolysis increases the hypoxic and acidic conditions in the tumor microenvironment and then forms a positive feedback loop with glycolytic enzymes.

### 2.1. Upregulation of Glycolysis

In previous cancer research, scientists have regarded a variety of glycolytic enzymes as housekeeping genes, and their expression levels have been used as loading controls in experiments [9]. However, as we learn more about glycolysis and cancer metabolism, the expression levels of these metabolic genes are also being recognized as variables that drive metabolic events and cancer phenotypes (Figure 2A). Unfortunately, current glycolytic enzymes cannot be fully studies in BC/UTUC. Although abnormal changes have received attention, some observations require more research and samples. Especially for UTUC, there is still a lack of clinical data (e.g., The Cancer Genome Atlas, TCGA) that can be analyzed, so this article is limited and focuses on BC. 

#### 2.1.1. GLUT

Increased glucose uptake results in increased glycolytic flux by GLUTs. The GLUT family includes 14 members, and GLUT1 expression has been associated with the grade of malignancy in non-muscle-invasive UC and muscle-invasive UC. In addition, GLUT1 is an independent prognostic marker for survival in BC patients treated with radiotherapy [10]. GLUT3 was also found to be upregulated in BC tumors [11]. Targeting GLUT1 and GLUT3 increases the sensitivity of BC cells to cisplatin and inhibits their proliferation, respectively [12,13].

#### 2.1.2. HK

In glycolysis, the first step is the phosphorylation of glucose by HK to form glucose-6-phosphate (G6P). In mammalian tissue, HKs have four isoforms: HK I, II, III, and IV (glucokinase) [14,15,16]. HK II expression maintains the malignant state in cancer, and it needs other protein partners to promote cancer progression, including GLUT, voltage-dependent anion channel (VDAC), ATP synthase, and adenine nucleotide translocator [17]. In BC, targeting HK2 suppresses BC progression [18]. Moreover, single-cell sequencing has been used to assess the activity of hexokinase 2 in urine for the non-invasive diagnosis and screening of BC [19].

#### 2.1.3. GPI

GPI carries out the second step of catalytic glycolysis. In BC, GPI and other glycolytic enzymes are upregulated during cancer progression [20]. Overexpression of GPI has also been detected in other cancer types [21].

#### 2.1.4. PFKM

Phosphofructokinase (PFK), which is a kinase enzyme that transfers a phosphoryl group to Fructose 6-phosphate (F6P) from adenosine triphosphate (ATP), is involved in an important reaction in the glycolysis pathway. Recently, the TCGA program recruited a large number of primary BC tissues and quantified the expression level of each gene through RNA-sequence approaches. Therefore, through the TCGA profiles, most of the PFK family (PFKL, PFKM, PFKP, PFKFB1, PFKFB2, PFKFB3, and PFKFB4) genes are amplified and upregulated in BC patients. Moreover, treatment with a PFK inhibitor (2,5-anhydro-d-glucitol-6-phosphate) significantly reduced glycolysis and inhibited the cell growth and invasion ability of BC cells [22].

#### 2.1.5. ALDO

In the fourth step, aldolase cleaves Fructose 1,6-bisphosphate (F1,6BP) into Glyceraldehyde 3-phosphate (GADP) and Dihydroxyacetone phosphate (DHAP). Aldolase family members are enzymes involved in the fourth step of the glycolysis process and include ALDOA, ALDOB, and ALDOC. Overexpression of ALDOA promotes cell growth, increases the colony formation rate, and enhances invasion ability in BC cells. ALDOA was also observed to modulate the expression of EMT marker, epidermal growth factor receptor (EGFR), mitogen-activated protein kinase (MAPK), and AKT serine/threonine kinase (AKT) phosphorylation levels in BC cells. ALDOA mRNA expression was associated with the stage of cancer and survival in patients with BC [23]. 

#### 2.1.6. TPI

TPI catalyzes the reversible interconversion between d-glyceraldehyde-3-phosphate (G3P/GA3P) and DHAP [24]. G3P is mainly used as a glycolysis intermediate, and DHAP is a precursor for fatty acid biosynthesis. TPI is regarded as a switch for conversion [25].

#### 2.1.7. GAPDH

GAPDH is involved in the glycolysis process and has been regarded as a housekeeping gene in previous studies. A large number of studies have used it as an internal control for DNA, RNA, or protein levels [26]. However, GAPDH can regulate glycolysis, gene transcription, apoptosis, and reactive oxygen species (ROS) response [27]. In BC, GAPDH is mentioned because of its abnormal performance, but there is still a lack of relevant research on UTUC.

#### 2.1.8. PGK

All members of the PGK family convert 1,3-diphosphoglycerate to 3-phosphoglycerate. Plant genomes contain up to three PGK genes, whereas mammals have two copies and most bacteria have one [28]. Tumor cells have been found to secrete PGK, which reduces the disulfide bonds in serine proteases and plasmin, thereby releasing angiostatin, a tumor vasculature inhibitor, and promoting angiogenesis [29,30]. PGK2 is transcribed from PGK1 via retrotransposition [31]. The gene does not contain introns and is specifically expressed in the testis. Similar to the situation of GAPDH, some glycolytic enzymes (PGK, PGAM, ENO, and PKM) still lack evidence and sufficient profiles to confirm their expression levels in clinical specimens. In contrast, using relatively sufficient BC datasets, we determined that these candidates have signs of overexpression in the tumor (Figure 2A).

#### 2.1.9. PGAM

In the eighth step of glycolysis, enzymes in the PGAM family convert 3-phosphoglycerate (3PG) into 2-phosphoglycerate (2PG) through the transfer of a phosphate group. There are five isoforms in the PGAM family, including PGAM1, PGAM2, PGAM4, and PGAM5. The fifth is PGAM3, which is a pseudogene. These PGAMs are similar to other glycolytic enzymes in different organs, and more importantly, their intracellular distribution is different. According to a previous review, PGAM1 and PGAM4 are located in the nucleoplasm, PGAM2 is located in the cytosol, and PGAM5 is located in the mitochondria [32]. Through the comparison of the human protein atlas website, PGAM1 is mainly expressed in BC, while PGAM5 is upregulated in UTUC. Such differences still need to rely on further research to identify and clarify.

#### 2.1.10. ENO

Enolase catalyzes the conversion of 2-phosphoglycerate to phosphoenolpyruvate (PEP), which consists of three family members: α-enolase (ENO1), γ-enolase (ENO2), and β-enolase (ENO3). Similar to other glycolytic enzymes, the enolase family behaves differently in most tissues. Unlike ENO2 (neuronal cells) and ENO3 (muscles), ENO1 has been found to be overexpressed in many cancers [33]. In various solid tumors, ENO1 has also been reported to increase glycolytic turnover and promote tumorigenesis. Moreover, ENO1 can divide a full-length canonical α-enolase into a truncated version, called Myc promoter-binding protein 1 (MBP1) [34]. In addition, their prognostic/diagnostic value in cancer has been confirmed [35]. In urothelial cancer, high expression of ENO1 is positively correlated with poor prognosis, ENO2 is negatively correlated, and ENO3 has no performance in tissue slides [36,37].

#### 2.1.11. PKM

Pyruvate Kinase isozymes M1/M2 (PKM1/PKM2) are responsible for the conversion of phosphoenolpyruvate (PEP) to pyruvate during glycolysis [38]. The family is composed of PKM1 and PKM2, which are differentially distributed, localized, and modified in cells. PKM1 is ubiquitously expressed in organs, and PKM2 is upregulated in tumorigenesis in various cancers, including urothelial cancer [39]. They consist of 12 exons and 11 introns, but changes in splicing lead to different protein molecular weights and structures [40]. Although some studies mention that PKM1-PKM2 undergoes a transition or shift in tumorigenesis, this remains to be thoroughly investigated. More research is needed to detect the consequences of PKM2 overexpression and activation [41,42]. Two isomeric forms (dimers and tetramers) of PKM2 have been found to be important factors in glycolysis [43]. The dimeric form of PKM2 is involved in fermentation to produce lactic acid, especially in cancer cells, while the tetrameric form continues to activate aerobic glycolysis and subsequent oxidative phosphorylation. These two forms correspond to the tissue type, PKM enzyme activity, and local tumor environment [44].

**Table 1 ijms-22-10612-t001:** Enzymes of glycolysis pathway.

Enzyme	Gene	Function in Cancer	Reference
Hexokinase	HK	cancer progression	[14]
Glucose-6-Phosphate Isomerase	GPI (PGI)	cancer progression	[20]
Phosphofructokinase 1	PFKM	cell growth and invasion ability	[15]
Aldolase, Fructose-Bisphosphate A	ALDOA	cell growth, colony formation rate, and invasion ability	[16]
Triosephosphate Isomerase 1	TPI1	NA	
Glyceraldehyde-3-Phosphate Dehydrogenase	GAPDH	gene transcription, apoptosis, and ROS response	[18]
Phosphoglycerate kinase	PGK	angiogenesis	[29,30]
Phosphoglycerate mutase	PGM	NA	
Enolase, phosphopyruvate hydratase	ENO	prognostic/diagnostic value in cancer has been confirmed	[35]
Pyruvate kinase	PKM2	tumorigenesis	[29,30]

### 2.2. Metabolic Reprogramming, Tumor Microenvironment, and Consequences

Aberrant glycolysis not only affects glucose-related products and intermediates but also triggers a series of critical cascade reactions, particularly metabolic reprogramming, activation of oncogenic pathways, and the promotion of various cancer phenotypes. Metabolic reprogramming occurs in the tumor microenvironment (TME) and influences the survival and function of tumor and immune cells. Their influence is not limited to changes in glucose. High levels of glucose cause excessive lactate formation and secretion into the extracellular space. The accumulation of extracellular lactate may create a TME that is beneficial for the malignant phenotype in cancer cells, including migration and angiogenesis, and it may also impair the immune system, allowing the tumor to evade immune detection [45]. Wang et al. observed that glycolysis-related genes (LDHA, TPI, and PGK1) and glutaminolysis-related genes (SLC1A5, GLS, and GLUD1) were elevated in BC and the surrounding T cells, reflecting the immune response. On the contrary, the dependence on oxidative phosphorylation (OXPHOs) is greatly reduced [46].

Afonso et al. described the relationship between glycolysis and immunotherapy. They claimed that competitive glucose metabolism is the goal of promoting immunotherapy for BC [47]. As mentioned above, the metabolic phenotype of BC cells is heterogeneous and differs from its counterparts in normal tissues because cancer cells have a strong demand for nutrients, especially glucose. There is an overlapping metabolic phenotype between cancer cells and activated T cells, leading to metabolic competition, limiting the availability of nutrients, increasing microenvironmental acidosis, and impairing the immune function of T cells. Numerous inhibitors of glucose metabolism have been shown to be effective in eliminating cancer cells that overexpress glycolysis-related targets, including silibinin, 2-deoxy-d-glucose, PFK158, gossypol, dichloroacetate, quercetin, metformin, and phenformin [48,49,50,51,52]. Integrating the targeting of BC metabolism into immunotherapy design seems to be a rational approach to improve the therapeutic efficacy of immune checkpoint inhibitors (Atezolizumab, Nivolumab, Duvalumab, Avelumab, and Pembrolizumab) [53,54,55,56,57,58,59].

## 3. Potential Signaling Pathways and Key Drivers of Glycolytic Enzymes

Glycolysis interacts with several carcinogenic pathways. These signals drive the expression levels of glycolytic enzymes through potential transcription factors, RNA interference, and modifications, forming a positive feedback loop.

### 3.1. Signaling Pathway 

The Phosphoinositide 3-kinase (PI3K)/protein kinase B (AKT)/mammalian target of rapamycin (mTOR) pathway is a common metabolic pathway that induces and regulates the Warburg effect [60,61]. In BC, this axis drives the increase in the glycolysis rate and is associated with the overexpression of several core glycolysis-related genes, such as GLUT1, HK1/2, and LDHA [8,62,63,64].

In addition, HLA-F locus adjacent transcript 10 (FAT10) promotes cell growth and mediates HK2 expression through the EGFR/AKT pathway in BC cells. It was also shown that the upregulated expression of HK2 correlated with FAT10 in BC patients [65]. On the other hand, PKM2 modulated the expression of SREBP-1c via the AKT/mTOR signaling pathway, which suppressed the transcription of fatty acid synthase (FASN, a major lipogenic gene), leading to reduced tumor growth. The PKM2 inhibitor (shikonin) was also shown to reduce cell proliferation and colony-forming abilities in a BC cell line [66].

### 3.2. Regulators

#### 3.2.1. HIF-1α

Hypoxia-inducible factor 1 alpha (HIF-1α) is one of the most well-known transcription factors, and its function is closely related to its role in metabolic reprogramming. Modification of HIF-1α in tumorigenesis has been identified in many studies. For example, Zhao et al. investigated the relationship between steroid receptor coactivator-3 (SRC-3) and HIF-1α in BC. They reported that SRC-3 can target the promoter region of HIF-1α and, as a result, accelerate the development of hypoxic conditions and anaerobic glycolysis through HIF-1α and its downstream target glycolytic enzymes [67]. Similarly, JMJD1A is a histone demethylase that specifically demethylates H3K9me 1/2. In a BC model, JMJD1A cooperated with HIF-1α and then upregulated various downstream genes, including some glycolytic enzymes, such as HK2, LDHA, and PGK1 [68].

HIF-1α is a well-known transcription factor and is activated in tumorigenesis. A large amount of research has validated that HIF-1α is stabilized, extended, and then translocated to the nucleus, where it affects downstream pathways. In addition to its ability to directly target the hypoxia response element (HRE) in many glycolytic enzymes, HIF-1α also participates in microRNA- and RNA-binding mechanisms to regulate glycolytic enzyme RNA levels [69,70,71]. Wang confirmed that Aly/REF export factor (ALYREF) enhances RNA stability in the PKM2 3′UTR region, and the HIF-1α/ ALYREF/PKM2 axis is formed to regulate glycolysis and proliferation in BC [72].

#### 3.2.2. TP53

TP53 is a well-known tumor suppressor, but it frequently undergoes mutation in tumorigenesis in various cancers. Once the loss of function of TP53 occurs, DNA damage, apoptosis, and repair mechanisms are affected [73,74]. At the same time, TP53 acts as a transcription factor that directly binds to the E3 ubiquitin ligase, which facilitates the degradation and stability of nuclear proteins [75]. 

#### 3.2.3. Insulin

Insulin acts as an additional factor and amplifies the signal transduction of glycolysis. It has been observed that certain hormones/growth factors favor insulin secretion in some cancer types and increase the risk of cancer development. In particular, when insulin is sufficient, GLUT4 acts as an insulin-responsive glucose transporter that continues to initiate the glycolysis mechanism. Although insulin concentration may not be a major risk factor for BC [76], it is still important to mention that high glucose (55 mM) and insulin (50 U) can promote the cell cycle progression and oncogenic signaling activation of bladder epithelial cells [77].

### 3.3. Modifications

In addition to genetic alterations that affect RNA expression, epigenetic modification is also an important mechanism for regulating expression levels. Downregulation of miR-125b-5p in BC tissues and cancer cell lines and low miR-125b-5p expression were associated with decreased 5-year survival in patients. Furthermore, targeting HK2 with miR-125b-5p inhibited BC cell progression [18]. 

Hypomethylation of the PGK1 promoter has been observed in several cancer subtypes. Through TCGA pan-cancer profiles, Shao et al. described the PGK1 promoter hypomethylation status in bladder cancer and others [78]. Although their study did not confirm that the hypomethylation status or expression level of PGK1 can be used as an indicator of different stages, they validated that PGK1-related events can serve as independent prognostic factors and are associated with poor survival. In addition, they also reported that the S203 site of PGK1 can be phosphorylated by ERK at the tumor site [79].

### 3.4. Protein–Protein Interactions

The non-enzymatic functions of many glycolytic enzymes have been identified in diseases and cancers [80]. These mechanisms can occur through direct protein–protein interactions rather than metabolites and metabolism-related pathways. These changes do not replace regular functions but support the mechanism of disease occurrence. For example, ALDOA-γ-actin and PGAM1-ACTA2 regulate the cell cytoskeleton and mobility, and PGK1-HTATSF1 promotes metastasis [81,82,83]. Researchers are working to produce accurate protein crystal structures and identify high-affinity binding partners of other glycolytic enzymes. Liu et al. focused on the C-terminal region of PGAM1 and validated the key residues in the PGAM catalytic cycle [84]. Zhai’s team and Butera’s group discovered that nuclear TP53–GAPDH forms a complex that supports glycolysis and drug resistance [85,86]. GPI is genetically identical to autocrine motility factor (AMF), a 55 kDa extracellular cytokine that can accelerate the migration of cancer cells [87].

### 3.5. Translocations

Several enzymes have been explored to characterize their translocation and their effects on comprehensive phenotypes in cancer. (1) When insulin-responsive glucose transporter, also known as GLUT4, receives the insulin signal, it is moved from GLUT4 storage vesicles to the cell membrane to increase glycolytic transport activity [88]. (2) Responding to the metabolic stress in cancer cells, HK2 is transferred to the outer mitochondrial membrane and improves metabolic capacity [89]. (3) The subcellular distribution of ALDOA is associated with survival signals and proliferation, and the nuclear localization of ALDOA responds to metabolic conditions [90]. (4) GAPDH has been confirmed to undergo nuclear translocation, let can induce apoptosis, and AKT2 can inhibit these events [91]. (5) Phosphorylation of PGK1 (S203) leads to mitochondrial translocation, which then decreases the TCA cycle and ROS production and increases pyruvate and lactate [79]. (6) A different form of ENO1, α-enolase, is localized in the cytoplasm, but MBP1 is preferentially trafficked to the nucleus for further repression of the transitional activity of c-Myc [92]. (7) PKM2 can exist as a tetramer to execute its enzyme function or as a dimer to carry out its non-glycolytic function in the nucleus [34]. Although these issues may be rarely mentioned and investigated in urothelial cancer, they are still important topics that need further analysis.

### 3.6. Non-Coding RNAs

In each cancer type, long non-coding RNAs are also involved in regulating the activity and metabolic effects of glycolytic enzymes [93]. Duan et al. observed that the lncRNA known as cancer susceptibility candidate 8 (CASC8) is reduced in the advanced stage of BC. Loss of function of CASC8 interferes with the binding of FGFR1, which in turn interferes with the phosphorylation state of LDHA [94]. On the other hand, urothelial cancer-associated 1 (UCA1) lncRNA upregulates HK2 expression and promotes glycolysis through the mTOR–STAT3 axis [95]. Similarly, some studies indicate that miR-143 mediates mTOR activity and HK2 expression [96].

Using two-dimensional polyacrylamide gel electrophoresis (2-DE)-based proteomics approaches, Peng et al. observed that PGAM1 was significantly upregulated in BC related to histological grade and poor differentiation. Dyrskjot and his group also claimed that miR-483-3p can target PGAM1 and PGAM4 [97,98]. In BC, miR-26a-5p and miR-26b-5p were determined to have tumor-suppressive effects, such as inhibiting cell aggressiveness, and glycolysis was proposed as one of the canonical pathways with potential involvement, along with downstream targets, such as HK2, ENO3, PGAM1, and PGAM4 [99]. Moreover, miR-21 was reported to reduce aerobic glycolysis in a BC cell line by increasing PTEN expression, decreasing AKT phosphorylation status, and inactivating mTOR. Furthermore, regulation of several glycolytic genes (GLUT1, GLUT3, HK1, HK2, LDHA, LDHB, and PKM) and events (glucose uptake and lactate production) was observed [100]. 

### 3.7. Others

Interestingly, Juwita et al. claimed that in two BC cell lines (RT4 and MGH), GAPDH was induced by Bacillus Calmette–Guerin, which is the standard therapy for non-muscle-invasive BC [101]. Although the specific mechanisms are still unclear, GAPDH is no longer suitable as a housekeeping gene for BC, as is in the case of gastric cancer [102]. According to Wu et al.’s report, esculetin can inhibit glycolysis by binding GPI and PGK2 in cancer cells, but the detailed mechanism remains to be explored [103].

## 4. Various Effects of Glycolytic Enzymes on Phenotypes of UC

### 4.1. Proliferation

The overexpression of most glycolytic enzymes has been revealed to enhance the proliferation ability of cancer cells by accelerating the efficiency and yield of glycolysis. In particular, the PI3K/Akt signaling pathway is a typical example of metabolic reprogramming/carcinogenic signal crosstalk. However, there are not many reports on the detailed molecular mechanism, Ji et al. reported that ENO1 promotes the growth and proliferation of BC cells through β-catenin and its downstream target cyclin D1 [104]. Xie et al. investigated the alternative splicing of PKM by PTBP1 regulated in BC. They confirmed that PKM2 is the essential factor in cancer proliferation. Although PTBP1 has been deprived, PKM2 remains and promotes the proliferation of BC [105].

### 4.2. Metastasis

Yan et al. reported that RNA-binding motif protein X-linked (RBMX) regulated cell proliferation, colony formation, and metastatic ability in metastatic BC [106]. They observed that RBMX competitively inhibited the combination of the RGG motif in hnRNP A1 and the sequence flanking PKM exon 9, leading to the formation of lower PKM2 and higher PKM1 levels, which attenuated the tumorigenicity and progression of metastatic BC. On the other hand, Li et al. observed that ALDOA was partially positively associated with the EMT phenotype to regulate the invasion of BC cells. Among non-invasive or invasive BC cells, EMT has been identified as an important issue and reflects cancer metastasis. ALDOA is associated with EGF/EGFR, MAPK/ERK, and PI3K/Akt pathways to reduce E-cadherin activity. At the same time, N-cadherin, vimentin, and mesenchymal markers were upregulated [23].

### 4.3. Drug Resistance

In urothelial cancer, platinum-based drugs such as cisplatin are included in the standard regimens for clinical patients. However, drug resistance often occurs in clinical cases and cell lines. Recently, T24 cells resistant to cisplatin (40 μM) (T24CDDPR) were established and compared with cisplatin-naïve (T24) cells through two-dimensional gel electrophoresis and LC-MASS. PGK1 was 1 of 25 differently overexpressed proteins (>1.5-fold change) in T24CDDPR cells compared with T24 cells [107]. Wang’s group claimed that, in addition to platinum-based compounds, shikonin can specifically bind to PKM2, but PKM1 does not. This interaction mechanism reduces the resistance of BC cells to cisplatin. The combined use of cisplatin and shikonin has a synergistic effect, increasing the rate of apoptosis and autophagy of cell lines. Moreover, they observed this regulation through the PKM2 enzyme-independent pathway, and therefore, it did not induce metabolic reprogramming [108].

## 5. Prognostic and Diagnostic Value of Glycolytic Enzymes for UC

For urothelial cancer, urine is a reliable source for the non-invasive detection of appropriate biomarkers. Various studies have analyzed changes in the urine of cancer patients compared with healthy subjects. Through high-throughput screening and an omics approach, several glycolytic enzymes were identified in these events. Liu et al. observed that PKM2 (0–1000 U/mL, *n* = 50) was highly expressed in the urine of cancer patients compared with the normal group (0 U/mL, *n* = 10). After evaluation, they determined that the ROC curve was 0.9, with 82% sensitivity and 100% specificity for BC [109]. In addition, there have been many studies on the prognosis/diagnosis of PKM2 in urothelial cancer, provided that PKM2 expression levels in the tumor and normal tissue can be defined. Physiological mechanisms such as localization and phosphorylation are easy to detect. Combined with existing clinicopathological factors, PKM2 has been revealed in various aspects of tumor pathology.

On the other hand, using 2-DE and MALDI-mass spectrometry, Zhou et al. reported that PKM2 is also overexpressed in urothelial cancer [110]. From low-grade papillary urothelial cancer to high-grade invasive urothelial cancer, the expression level of PKM2 increased, and translocation from the cytoplasm to the nuclear region occurred. In addition, the phosphorylation status of PKM2 increased at the same time. There was no significant change in PKM1 in the urothelial cancer cell panel. Huang et al. demonstrated that the RNA and protein levels of PKM2 are upregulated in BC. They collected 215 cases and calculated the relationship between PKM2 and clinical parameters. PKM2 expression was correlated with poor disease-free survival and served as an independent prognostic factor in multivariate analysis (HR = 3.5, *p* < 0.001) [111].

However, Lu et al. confirmed that the PGK1 protein level has several connections with clinical parameters of gallbladder cancer (GBC). In their GBC cohort (*n* = 95), the expression of PGK1 in GBC was significantly reduced compared with non-cancerous gallbladder mucosa. Through Kaplan–Meier analysis and univariate/multivariate approaches, PGK1 protein was correlated with a good survival rate and independent value of GBC. Moreover, PGK1 was stronger in GBC than TNM or had a comprehensive synergistic effect with TNM [112]. Most glycolytic genes were overexpressed and were associated with poor survival rates. PFKFB4 has been suggested as a potentially suitable prognostic marker in non-muscle-invasive BC (HR = 2.026 (1.177–3.488), *p* = 0.011) [113]. The prognostic value of GLUT1 has been revealed in BC, especially at a low stage and low grade [114,115]. These findings indicate that the upregulation of glycolytic enzymes is not a conclusion and consequence and can be further used to formulate clinical strategies and monitoring targets (Figure 2).

## 6. Omics Datasets Available in UC

To analyze the copy number alteration (CNA) events, mutation status, or expression level of each glycolytic enzyme in BC, multi-omics data have been established and updated over the past decade. These clinical cases have been collected and analyzed through large-scale screening by research institutions. From online websites and tools, normalized configuration files can be easily mined to obtain relevant data (Table 2). These data have been generated by microarray chips, RNA sequencing, genomics, epigenetic alterations, and proteomics. In addition to the above-mentioned clinical population, databases have also been established based on cell/animal experiments. These profiles can be divided into several types: (1) built models of overexpression or knockdown, (2) comparison of drug or carcinogen exposure, and (3) analysis based on specific phenotypes or signals.

As a practical application, Kim et al. detected FGFR3 alterations and analyzed the immune response in metastatic urothelial cancer (GSE176307). Phung et al. and Therkilden et al. used formalin-fixed paraffin-embedded tissue from a single Lynch Syndrome patient (GSE146670 and GSE104922) and studied changes across time, tissues, and space [3]. Sjodahl made some array chips to examine differentially expressed molecules in the early stage of advanced urothelial cancer (GSE128959) [116]. There are also several omics data profiles based on drug resistance design, such as GSE122358, GSE112973 [117], GSE98096 [118], GSE92651 [119], and GSE58624 [120].

Lee et al. identified the BC cell line J82 and its derived xenograft tumors, MDXC1 (GSE156348) [121]. In this study, they found that some tyrosine kinases may be potential treatment strategies. Lee et al. and Chen et al. used an RNA sequencing approach to distinguish normal/tumor pairs in patients with urothelial cancer (GSE159824 and GSE133624) [122,123]. Many scientists have generated their data from patient-derived xenografts (PDXs) and organoids (GSE155007 and GSE134292) [124]. Such research also includes GSE77952 [125], GSE125286, GSE111933 [126], GSE68020, GSE129441 [127], and GSE103990 [128].

Zavadil’s group performed a series of dose/time-dependent aristolochic acid exposure in patients and normal tissues. In these datasets, mRNA, microRNA, and urine sample data were included (GSE166716, 166,907, 166,909, and 166,912). Similarly, Jou et al. analyzed cyproheptadine and epigenetic modification in urothelial cancer (GSE160703) [129]. Additional studies have built models by adding medications, including GSE90023 [130], GSE45385, GSE74478 [131], GSE102170 [132], and GSE103928. These datasets may not be designed for metabolic changes or events, but comprehensive physical data can bring us closer to the clinical perspective and can increase our understanding of the dynamics of all metabolism-related genes and pathways, which are more useful in practice.

**Table 2 ijms-22-10612-t002:** Available studies for visualization and analysis of urothelial cancer (bladder/upper tract). All data and their corresponding clinical parameters were obtained through whole-exome or RNA sequencing approaches. These clinical cohorts were collected from the cBioportal website and previous studies. Please refer to the original references or cBioportal website for the definition of subtypes.

Source	Year	Cases	Reference
**Bladder cancer (BC)**			
MSK/TCGA	2020	476	NA
MSKCC	2014	109	[133]
MSKCC	2013	97	[134]
MSKCC	2016	34	[135]
TCGA	2017	413	[2]
BGI	2013	99	[136]
DFCI/MSKCC	2014	50	[137]
TCGA	2016	413	NA
TCGA	2014	131	[138]
TCGA	2018	411	[139,140,141,142]
MSK	2017	105	[143]
Cornell/Trento	2016	72	[144]
**Upper tract urothelial cancer (UTUC)**			
MSK	2015	85	[145]
Cornell/Baylor/MDACC	2019	47	[7]
IGBMC	2021	30	[146]
MSK	2020	119	[147]
MSK	2020	78	[147]

MSKCC: Memorial Sloan Kettering Cancer Center; TCGA: The Cancer Genome Atlas; IGBMC: Institute of Genetics and of Molecular and Cellular Biology; BGI: Beijing Genomics Institute.

## 7. Current Combinations and Clinical Trials Based on Glycolytic Inhibitors

### Potential Compounds/Strategies

For targeted glycolytic enzymes and metabolic reprogramming in tumorigenesis, reordering or interfering with intermediate production may have a dramatic impact. Previous studies have found that knocking down PGAM1 to accumulate 3-PG and reduce 2-PG production could inhibit glycolysis and the oxidative pentose phosphate pathway in BC cells [148]. Apoptosis and a decreased proliferation rate have also been induced in cancer cells in vitro and in vivo. In addition, HKB99, an allosteric inhibitor of PGAM1, was developed to block the interaction between PGAM1 and ACTA2 through structure-based optimization [82]. In this study, they compared the lead compound PGMI-004A and several candidate compounds (MJE3, EGCG, and 12r) with HKB99 and found that HKB99 had the most significant effect on lung cancer cells [149,150,151]. Although there is no clear evidence for HKB99 in urothelial cancer, some references indicate that it can be applied and tested.

In the past, inhibitors have been developed to block or interfere with enzymes involved in glycolysis. Several inhibitors are also used in urothelial cancer (Table 3). Choi summarized the anti-cancer effect of phloretin [2′,4′,6′-trihydroxyphenyl]-propiophenone]. In a BC study, phloretin was found to inhibit glucose uptake by reducing the expression of glucose transporters [152,153]. Quercetin has been identified as having cytotoxic and genotoxic effects on human BC cells [154], and it can induce BC apoptosis through the AMPK axis [155]. WZB117 uses non-specific inhibition to reduce the transport of glucose transporters, thereby slowing the rate of glycolysis. According to reports, ritonavir and indinavir produce similar trends to WZB117 [156]. Some drugs are used alone or in combination with existing chemotherapeutics to improve their efficiency against glycolytic enzymes. These candidate drugs include 3-bromopyruvate, 2-deoxyglucose, 3PO, 2-phospho-d-glyceric acid, sodium fluoride, and benzo[a]pyrene [48,157,158,159].

The role of certain compounds is to alter the processes of metabolic reprogramming and several branches of production. The consequence of these drugs is not the regulated expression of a single gene but a large-scale metabolic event. This also means that the use of the above drugs has limitations and involves physiological considerations. In addition, some valuable potential inhibitors (d-fructose-6-phosphate, 4-deoxy-d-glucose, iodoacetate, tubercidin, mellitic acid, 3-aminoenolpyruvate 2-phosphate, pyridoxal-5’-phosphate, oxamic acid, and NHI-1) can be administered to cells and animal models for further studies of urothelial cancer. 

## 8. Future Prospects

In this article, we describe the pathways and events of glycolytic enzymes involved in urothelial cancer. Although there are many studies on multiple genes in other cancer types, UTUC and BC are characterized by specific genetic alterations and features. This does not mean that previous studies on our topic can necessarily be replicated, and more research is needed to confirm their results and clarify their limitations.

Through the large-scale screening of urothelial cancer and the establishment of omics databases, it has been reported that the RNA or protein levels of various glycolytic enzymes may be abnormally expressed, and the enzymes may be dysfunctional. However, with the confirmation of genetic modification events and the development of specialized inhibitors, glycolytic enzymes need to be reevaluated not as independent events but to discover their additional clinical value. Previously, an FGFR inhibitor, Debio 1347 (CH5183284), was found to be a possible resistance mechanism [180]. The follow-ups—Infigratinib, Erdafitinib, Pemigatinib—are specially designed for FGFR members and have been applied to urothelial cancer [181]. If scientists determine the relationship between glycolytic enzymes and typical genetic changes, it may increase the response and synergistic effect in clinical practice.

In addition, past studies have recognized that glycolytic enzyme inhibitors can have strong side effects and may cause irreversible catalytic processes in the human body. Moreover, cancer cells have alternative feedback loops/branches to preserve specific deficient nutrients. There are many effects of glucose deprivation in cancer cells, but cells can be converted to use glutamate to maintain their behavior and function. Therefore, some glycolytic enzyme-related inhibitors have failed in animal models or clinical trials. On the contrary, new small-molecule drugs or peptides will be based on protein–protein interactions, and whether enzyme functions and metabolic events are affected will need to be carefully examined. These research directions will determine the application and translational medicine potential of glycolytic enzymes in urothelial cancer.

## Figures and Tables

**Figure 1 ijms-22-10612-f001:**
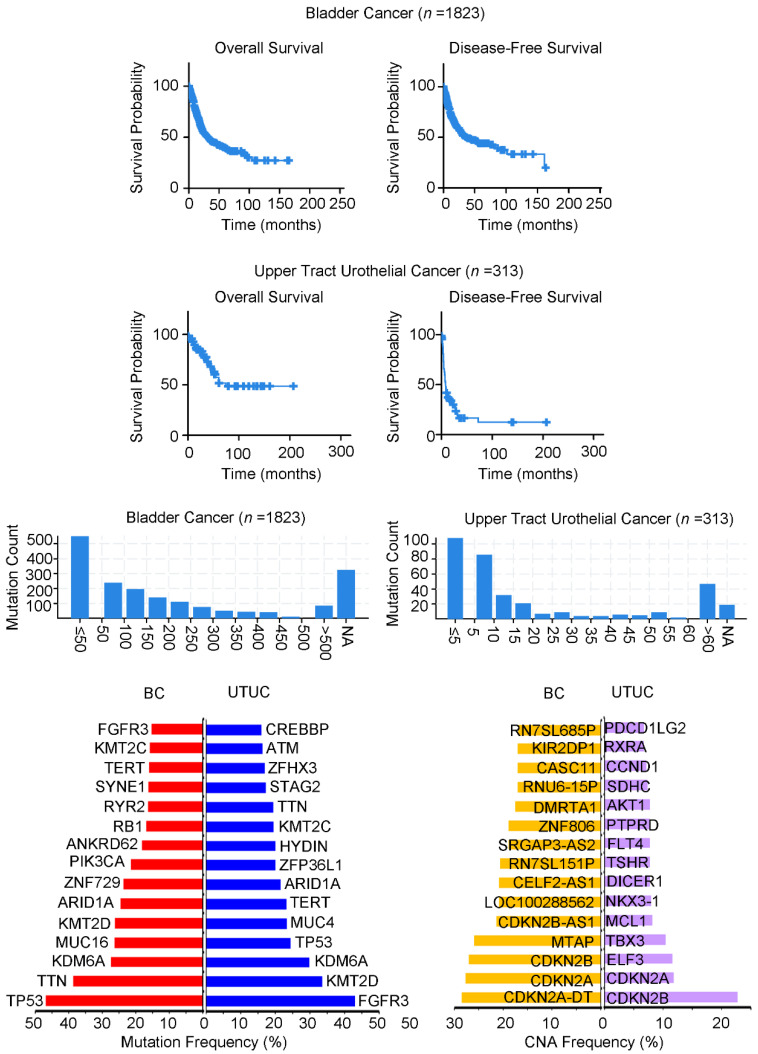
A meta-analysis of BC (*n* = 1823) and UTUC (*n* = 313) cohorts to reveal the survival rates, mutation frequency, and genetic alterations of clinical patients. The overall survival of BC (average 32 months) is worse than UTUC (average 60.79 months), but the disease-free survival rate of UTUC (average 8.5 months) is worse than BC (average 35.3 months). In addition, the TOP15 gene alterations and CNA changes of the two cancers are mostly different, with only a few repetitions but different frequencies. Genetic background and clinical parameters in urothelial cancer. All data were analyzed from the cBioportal website. CNA: Copy number alteration.

**Figure 2 ijms-22-10612-f002:**
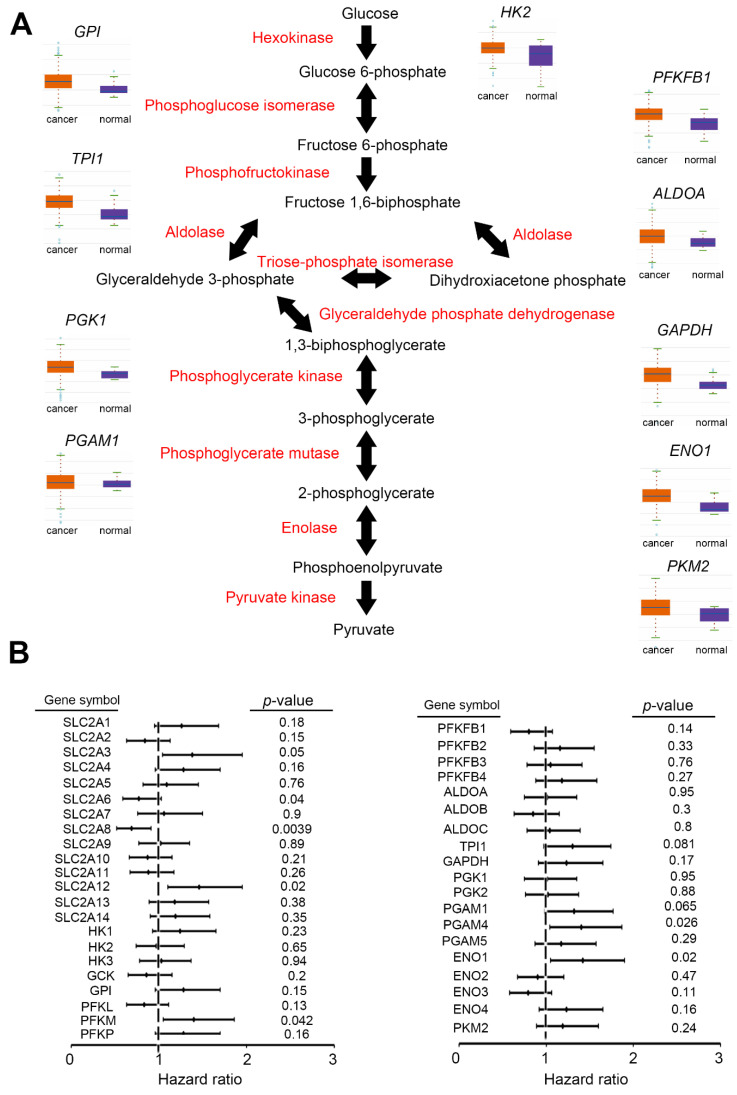
Glycolytic pathway was up-regulated in urothelial cancer. (**A**) Relative glycolytic enzymes mRNA levels in BC tissues. (**B**) Hazard ratio, *p*-value, and expression level of each glycolytic enzyme in urothelial cancer. All data were acquired from Kaplan–Meier Plotter, USCS Xena, and ENCORI websites.

**Table 3 ijms-22-10612-t003:** Development phase of glycolytic enzyme compounds in UC and their pre-clinical or clinical effects.

Inhibitor	Targets	Reference	Application in UC
Phloretin	GLUTs	[160,161]	[162]
Quercetin	GLUT1/PKM2/LDHA	[161,163]	[154,155]
STF31	GLUT1	[161]	[48]
WZB117	GLUT1	[161]	[164]
3-Bromopyruvate	HK	[165]	[48,157]
2-Deoxyglucose	HK	[166]	[48]
D-fructose-6-phate	GPI	[167]	NA
3PO	PFKFB	[168]	[48]
4-Deoxy-d-glucose	Aldolase	[81]	NA
2-phospho-d-glyceric acid	TPI	[169]	[158]
Iodoacetate	GAPDH	[170]	NA
Tubercidin	PGK	[171]	NA
Mellitic acid	PGAM	[167]	NA
Sodium fluoride	ENO	[172]	[173]
3-aminoenolpyruvate 2-phosphate	ENO	[169]	NA
Shikonin	PKM2	[174,175]	[176,177]
Pyridoxal-5′-phosphate	PKM2	[167]	NA
Oxamic acid	LDH	[178]	NA
NHI-1	LDHA	[179]	NA
Benzo[a]pyrene	Glycolysis/PPP	[159]	[159]

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
