# Peer review of "Current Development and Application of Anaerobic Glycolytic Enzymes in Urothelial Cancer"

_ijms, 2021, doi:10.3390/ijms221910612_

Round 1
Reviewer 1 Report
This is a narrative review on the involvement of the glycolytic pathway in UC. In the reviewer’s impression, accumulated evidence is too immature to review the therapeutic application and draw the conclusion.
The authors may want to address the following points.
1_ Chapter 2.1.: There seems to be no report on UC for several enzymes, which should be clarified.
2_ Table 3: The authors may want to describe the development phase of the compounds for UC and their pre-clinical or clinical effects.
Minor
Abstract: “urinary system tumors” should be corrected as UC.
Author Response
Reviewer 1:
This is a narrative review on the involvement of the glycolytic pathway in UC. In the reviewer’s impression, accumulated evidence is too immature to review the therapeutic application and draw the conclusion.
The authors may want to address the following points.
1_ Chapter 2.1.: There seems to be no report on UC for several enzymes, which should be clarified.
Answer: We thank the reviewer for useful suggestions. We added some description in the section. 2. Please find the lines 109~113, 165-167 and 175-179 in the revised manuscript. “Unfortunately, current glycolytic enzymes cannot be fully studies in UBC/UTUC. Although abnormal changes have received attention, some observations require more re-search and samples. Especially for UTUC, there is still a lack of clinical data (ex. The Cancer Genome Atlas, TCGA) that can be analyzed, so this article is limited and focuses on UBC.”
2_ Table 3: The authors may want to describe the development phase of the compounds for UC and their pre-clinical or clinical effects.
Answer: We thank the reviewer for the comment. We have edited the title in the Table 3. “Table 3. Development phase of glycolytic enzyme compounds in UC and their pre-clinical or clinical effects.”.
Minor
Abstract: “urinary system tumors” should be corrected as UC.
Answer: Thank you for your suggestion. We corrected these errors.
Reviewer 2 Report
This is a comprehensive review.
a bit long but as a comprehensive review I did not asked to shorten beside introduction.
Author Response
Reviewer 2:
This is a comprehensive review.
a bit long but as a comprehensive review I did not asked to shorten beside introduction.
Answer: We appreciate your comments and suggestions. Since UTUC and UBC have different genetic backgrounds, they may reflect different metabolic events. So we hope that readers can understand the detailed background of the two in the introduction part.